# Estimating the Parameters of Fitzhugh–Nagumo Neurons from Neural Spiking Data

**DOI:** 10.3390/brainsci9120364

**Published:** 2019-12-09

**Authors:** Resat Ozgur Doruk, Laila Abosharb

**Affiliations:** Department of Electrical and Electronics Engineering, Atılım University, Incek, Golbasi, 06836 Ankara, Turkey; lailaabusharb@yahoo.com

**Keywords:** neuron modeling, Fitzhugh–Nagumo Model, Poisson processes, inhomogeneous Poisson, neural spiking, maximum likelihood estimation

## Abstract

A theoretical and computational study on the estimation of the parameters of a single Fitzhugh–Nagumo model is presented. The difference of this work from a conventional system identification is that the measured data only consist of discrete and noisy neural spiking (spike times) data, which contain no amplitude information. The goal can be achieved by applying a maximum likelihood estimation approach where the likelihood function is derived from point process statistics. The firing rate of the neuron was assumed as a nonlinear map (logistic sigmoid) relating it to the membrane potential variable. The stimulus data were generated by a phased cosine Fourier series having fixed amplitude and frequency but a randomly shot phase (shot at each repeated trial). Various values of amplitude, stimulus component size, and sample size were applied to examine the effect of stimulus to the identification process. Results are presented in tabular and graphical forms, which also include statistical analysis (mean and standard deviation of the estimates). We also tested our model using realistic data from a previous research (H1 neurons of blowflies) and found that the estimates have a tendency to converge.

## 1. Introduction

Application of computational tools in neuroscience is an emerging field of research in the last 50 years. The Hodgkin–Huxley model [1] is a striking development in the field of theoretical and computational neuroscience. Here, the membrane potential and its bursting properties are modeled as a fourth-order nonlinear system. In addition to the membrane potential, it describes the behaviors of sodium and potassium ion channels. Its nonlinear properties make some researchers search for possibilities that yield simpler nonlinear differential equations. One such attempt is the second-order Morris–Lecar [2], which lumps the ion channel activation dynamics into a single recovery variable. It is still a conductance based model. Further simplifications involve complete elimination of physical parameters such as ion conductances. Two major examples are the second-order Fitzhugh–Nagumo [3,4] and the third-order Hindmarsh–Rose [5] models. These can model the pulses and bursts occurring in the membrane potential without the need of physical parameters like ion conductances. In addition, as in the case of Morris–Lecar models, the behaviors of ion channels are lumped into generic variables.

In the case that only the input output (stimulus/response) relationships are important, general neural network models can be a good choice. Some examples from literature are the static feed-forward models [6,7] and nonlinear recurrent dynamical neural network models [8,9]. The dynamical neural network models can be structured such that one may receive a membrane potential information (bursts can be explicitly recovered) or just the instantaneous firing rate as the output [10]. In addition, sometimes only the statistical properties of the stimulus/response pair is import and thus statistical black-box models are taken into account [11,12].

Regardless of the chosen model, stimulus/response data are required to obtain an accurate relationship. Depending on the experiment, these data may be continuous or discrete in nature. In the case of an in-vitro environment such as a patch clamp experiment, one may record a full time dependent profile of membrane potential. That allows computational biologists to perform an identification (parameter estimation) based on traditional minimum mean square estimation (MMSE) techniques. However, in an in-vivo experiment, it is very difficult to collect continuous data revealing exact (or in an acceptable range at least) membrane potential information. If a membrane potential micro electrode contacts a living neuron membrane, the resistive and capacitive properties of the electrode may alter the operation of the neuron. This is not desired as one will not model a realistically functioning neuron at the end of the identification process.

In [7,8], it is suggested that one can record the successive action potential timings if the electrodes are suitably placed in surroundings of the membrane. With that, one is able to form a neural spike train which has the discrete timings of the spikes (or of the action potential bursts). Of course, a spike train cannot have dedicated amplitude information. However, this does not mean that one is hopeless concerning model identification. In [13], it is suggested that neural spike timings largely obey Inhomogeneous Poisson Point Processes (IPPPs). Being aware of the fact that an IPPP can be approximated by a local Bernoulli process [14], it would be convenient to derive suitable likelihood functions and apply statistical parameter identification techniques on that.

In addition, previous research suggests that the transmitted neural information is not directly coded by the membrane potential level but rather vested in the firing rate [15], interspiking intervals (ISI) [16] or individual timings of the spikes [17]. Thus, training neuron models from discrete and stochastic spiking data is expected to be a beneficial approach to understand the computational features of our nervous system.

Concerning the application of statistical techniques based on point process likelihoods to neural modeling, there are a few research works in the related literature. The authors of [6,7] applied maximum a-posteriori estimation (MAP) technique to identification of the weights of a static feed-forward model of the auditory cortex of marmoset monkeys. The authors of [8,9] presented a computational study aiming at the estimation of the network parameters and time constants of a dynamical recurrent neural network model using point process maximum likelihood technique. The authors of [18] applied likelihood techniques to generate models for point process information coding. The authors of [19] trained a state space model from point process neural spiking data.

In a few research studies, Fitzhugh–Nagumo models are involved in stochastic neural spiking related studies. For example, the authors of [20] dealt with the interspike interval statistics when the original Fitzhugh–Nagumo model is modified to include noisy inputs. The number of small amplitude oscillations has a random nature and tend to have an asymptotically geometric distribution. Bashkirtseva et al. [21] studied the effect of stochastic dynamics represented by a standard Wiener process on the limit cycle behavior. In [22], the authors performed research on the hypoelliptic stochastic properties of Fitzhugh–Nagumo neurons. They studied the effect of those properties on the neural spiking behavior of Fitzhugh–Nagumo models. Finally, Zhang et al. [23] investigated the stochastic resonance occurring in the Fitzhugh–Nagumo models when trichotomous noise is present in the model. They found that, when the stimulus power is not sufficient to generate firing responses, trichotomous noise itself may trigger the firing.

In this research, we treated a conventional single Fitzhugh–Nagumo equation [3,4] as a computational model to form a theoretical stimulus/response relationship. We were interested in the algorithmic details of the modeling. Thus, we modified the original equation to provide firing rate output instead of the membrane potential. Based on the findings in [8,9,10], we mapped the firing rate and membrane potential of the neuron by a gained logistic sigmoid function. Sigmoid functions have a significance in neuron models as they are a feasible way of mapping the ion channel activation dynamics and membrane potential [1,2].

Although the output of our model is the neural firing rate, the responses from in vivo neurons are stochastic neural spike timings. To obtain representative data, we simulated the Fitzhugh–Nagumo neurons with a set of true reference parameters and then generated the spikes from the output firing rate by simulating an Inhomogeneous Poisson process on it.

The parameter estimation procedure was based on maximum likelihood method. Similar to that of Eden [14], the likelihood was derived from the local Bernoulli approximation of the inhomogeneous Poisson process. That depends on the individual spike timings rather than the mean firing rate (which is the case in Poisson distribution’s probability mass function).

The stimulus was modeled as a Fourier series in phased cosine form. This choice was made to investigate the performance of the estimation when the same stimulus as that in [8,9] was applied. In the computational framework of this research, the stimulus was applied for a duration 30 ms. This may be observed as a relatively short duration and it is chosen to speed up the computation. In some studies (e.g., [24,25,26]), one can infer that such short duration stimuli may be possible for fast spiking neurons.

In addition, fast spiking responses obtained from a single long random stimulus can be partitioned to segments of short duration such as 30 ms. Thus, the approach in this research can also be utilized in modeling studies that involves longer duration stimuli.

In addition to the computational features of this study, we also investigated the performance of our developments when the training data are taken from a realistic experiment. To achieve this goal, we used the data generated by de Ruyter and Bialek [27]. The data from this research have a 20 min recording of neural spiking responses obtained from H1 neurons of blowfly vision system against white noise random stimulus. The response was divided into segments of 500 ms and the developed algorithms were applied. Each 500 ms segment can be thought as an independent stimulus and its associated response.

## 2. Materials and Methods

### 2.1. FitzHugh-Nagumo Model

Fitzhugh–Nagumo (FN) model is a second-order polynomial nonlinear differential equation bearing two states representing the membrane potential (V) and a recovery variable (W), which lumps all ion channel related processes into one state. Mathematically, it can be represented as shown below [28]:(1)$$\begin{array}{c} \begin{array}{cc} \dot{V} & =V-dV^{3}-W+I\\ \dot{W} & =cV+a-bW \end{array} \end{array}$$

The above model has four parameters [a,b,c,d] determining its properties. In the original text associated with the FN models, the coefficient of the $V^{3}$ is MM1/3; however, in this work, we suppose that the coefficient of that cubic term is not constant and we assign a parameter *d* to it. In Equation (Equation 1), I represents the stimulus exciting the neuron. It can be thought of as an electric current.

In the introduction, we state that we need a relationship between the membrane potential representative variable *V* and the firing rate of our neuron. In addition, we also state that we can construct such a map by developing a nonlinear sigmoidal map as shown below:(2)$$r=\frac{F}{1+\exp(-V))}$$
where *r* is the firing rate of the neuron in ms${}^{-1}$ and *F* is the maximum firing rate parameter. Thus, one has five parameters to estimate and they can be vectorally expressed as:(3)θ=[a,b,c,d,F]

Thus, we can call $\hat{\theta }$ as the estimates of θ. In the application, we needed the true values of θ so that we coul generate the spikes that represent the collected data from a realistic experiment. These are available in Table 1.

### 2.2. Stimulus

The signal for stimulation was modeled using a phased cosine Fourier series as:(4)$$I=\sum_{n=1}^{N_{U}}A_{n}\cos\left(\omega_{n}t+\phi_{n}\right)$$
where $A_{n}$ represents the amplitude, $\omega_{n}=2\pi f_{0}n$ stands for the frequency of the *n*th Fourier component in MMrad/sec, and $\phi_{n}$ stands for the phase of the component in radians. The amplitude $A_{n}$ along with the base frequency $f_{0}$ (in Hz) were kept constant, whereas the phase $\phi_{n}$ was selected randomly from a uniform distribution in [−π,π] radians. The amplitude parameter $A_{n}$ was unchanged for all mode *n* and it was set as $A_{n}=A_{\max}$.

### 2.3. Neural Spiking and Point Processes

We state in the introduction that the neural spiking is a point process that largely obeys an Inhomogeneous Poisson Process (IPP). A basic Poisson process is characterized by an event rate λ and has an exponential probability mass function defined by:(5)$$Prob\left[N\left(t+\Delta t\right)-N\left(t\right)=k\right]=\frac{e^{-\lambda }\lambda^{k}}{k!}$$
where *k* is the number of events that occur in the interval $\left[t,t+\Delta t\right)$. In the simplest case, λ is constant in that interval. In neural operation, the process is much more complex and assuming a constant event rate is insufficient; thus, we refer to a time varying event rate, which is actually equivalent to the firing rate r(t) of the neuron (refer to Equation (Equation 2)). This yields an inhomogeneous Poisson point process with the event rate λ replaced by the mean firing rate defined by:(6)$$\lambda =\int_{t}^{t+\Delta t}r\left(\tau \right)d\tau$$

Now, the term *k* represents the spike count in the interval $\left[t,t+\Delta t\right)$, which is statistically related to the firing rate r(t); λ now represents the mean spike count for the firing rate r(t), which varies with time; and N(τ) stands for the cumulative total number of spikes up to time τ, thus making $N\left(t+\Delta t\right)-N\left(t\right)$ the spike count for the time interval $\left[t,t+\Delta t\right)$.

Now, let us take a spike train $\left(t_{1},t_{2},\ldots ,t_{K}\right)$ in the time interval (0,T). Here, $0\leq t_{1}\leq t_{2}\leq \ldots \leq t_{K}\leq T$, thus *t* and Δt become 0 and *T*. The spike train can be defined using a series of time stamps for *K* spikes. As a result, the likelihood density function related to any spike train $\left(t_{1},t_{2},\ldots ,t_{K}\right)$ is gained using an inhomogeneous Poisson process [14,30] in the following way:(7)$$p\left(t_{1},t_{2},\ldots ,t_{K}\right)=\exp\left(-\int_{0}^{T}r\left(t,x,\theta \right)dt\right)\prod_{k=1}^{K}r\left(t_{k},x,\theta \right)$$

The function reveals the likelihood of a given spike train $\left(t_{1},t_{2},\ldots ,t_{K}\right)$ to occur with the rate function $r\left(t,x,\theta \right)$, which obviously is relying mainly upon network parameters and the stimulus applied.

### 2.4. Maximum Likelihood Methods and Parameter Estimation

The parameters requiring assessment appear as a vector:(8)$$\theta =\left[\theta_{1},\ldots ,\theta_{5}\right]=\left[\hat{\theta_{1}},\ldots ,\hat{\theta_{5}}\right]$$
to cover all the parameters in Equation (Equation 3). The maximum probability here relies on the function proposed in Equation (Equation 7) and includes each spike timing as well. Estimation theory asserts that determining maximum probability is asymptotically effective and goes as far as the Cramér–Rao bound within the scope of large data. Therefore, for us to expand the probability function in Equation (Equation 7) to further cover settings with numerous spike trains initiated by numerous stimuli, a series of *M* stimuli should be assumed. Take the *m*th stimulus (m=1,…,M) to initiate a spike train containing $K_{m}$ spikes in the time window [0,T], and the spike timings are given by $S_{m}=\left(t_{1}^{\left(m\right)},t_{2}^{\left(m\right)},\ldots ,t_{K_{m}}^{\left(m\right)}\right)$. By Equation (Equation 7). According to Equation (Equation 7), the probability function for the spike train $S_{m}$ can be determined as:(9)$$p\left(S_{m}|\theta \right)=\exp\left(-\int_{0}^{T}r^{\left(m\right)}\;\left(t\right)dt\right)\prod_{k=1}^{K_{m}}r^{\left(m\right)}\;\left(t_{k}^{\left(m\right)}\right)$$
in which $r^{\left(m\right)}$ represents the firing rate due to the *m*th stimulus. Let us denote that the rate function $r^{\left(m\right)}$ entirely relies on the parameters related to neuron parameters θ and the stimulus. On the left-hand side of Equation (Equation 9), its reliance on the neuron parameters θ can be noted.

Supposing the stimulus and its elicited responses in each *m*th trial are independent, one can derive a joint likelihood function as:(10)$$L\left(S_{1},S_{2},\ldots ,S_{M}|\theta \right)=\prod_{m=1}^{M}p\left(S_{m}|\theta \right)$$

To improve its convexity, we can make use of natural logarithm and derive a log likelihood function as shown below:(11)$$l\left(S_{1},S_{2},\ldots ,S_{M}|\theta \right)=-\sum_{m=1}^{M}\int_{0}^{T}r^{\left(m\right)}\;\left(t\right)dt+\sum_{m=1}^{M}\sum_{k=1}^{K_{m}}\ln r^{\left(m\right)}\;\left(t_{k}^{\left(m\right)}\right)$$

Finally, the maximum likelihood estimates of the parameter vector θ is obtained by:(12)$${\hat{\theta }}_{ML}=\arg\max_{\theta }\left[l\left(S_{1},S_{2},\ldots ,S_{M}|\theta \right)\right]$$

### 2.5. Spike Generation for Data Collection

Since this study was of computational type and targeted the development of an algorithm to be applied in a realistic experiment, we needed a solid approach to generate a dataset to represent the output of a realistic experiment. In the current research, the data were a set of neural spike trains that bear the individual spike timings with no amplitude information. In addition, we also know that the neural spiking process largely obeys inhomogeneous Poisson statistics, thus we could achieve that goal by implementing a stable Poisson process simulation. In other words, we simulated an inhomogeneous Poisson process with r(t) as its event rate. There are several algorithms to simulate an inhomogeneous Poisson process. The local Bernoulli approximation [14], thinning [31], and time scale transformation [32] can be shown as examples.

If the time bin is sufficiently small (e.g., δt=10μs) such that only one spike is fitted, one can use local Bernoulli approximation to generate the neural spiking data very easily. The is also a reasonable choice when the neuron models are integrated by discrete solvers such as the Euler or Runge–Kutta method. One can see a summary of the related algorithm below [8]:Given the firing rate of a neuron as r(t).Find the probability of firing at time $t_{i}$ by evaluating $p_{i}=r\left(t_{i}\right)\delta t$ where δt is the integration interval. It should be a small real number such as 1 ms.Draw a random number $x_{rand}=U\left[0,1\right]$ that is uniformly distributed in the interval [0,1]. Here, *U* stands for a uniform distribution.If $p_{i}>x_{rand}$, fire a spike at $t=t_{i}$, else do nothing.Collect spikes as $S=[t_{1},\ldots ,t_{N_{s}}]$ where $N_{s}$ will be the total number of spikes collected from one simulation.

## 3. Application

In this section, we introduce a simulation-based approach to evaluate the parameters of a firing rate-based single Fitzhugh–Nagumo neuron model. The process in brief appears as follows:A single run of simulation lasted for $T_{f}=30$ ms.The stimulus amplitude $A_{\max}$ and base frequency $f_{0}$ were assigned prior to each trial *m*. The phase angles $\phi_{n}$ was assigned randomly, as defined in Section 2.2.The firing rate profile was obtained by integrating the FN model in Equation (Equation 1) for $T_{f}=30$ ms using a time bin of $\delta_{t}=10$μs. The integration was performed at the true values of the parameters in Table 1 to generate the actual firing rate information $r^{m}\left(t\right)$ of current *m*th trial.Using the approach presented in Section 2.5, the spike train $S_{m}$ of the *m*th trial was generated from the firing rate $r^{m}\left(t\right)$. The number of spikes was $K_{m}$ at the *m*th trial.The simulation was repeated $N_{it}$ times to collect several statistically independent spike trains, i.e., $m=1\ldots N_{it}$.The neural spiking data needed by Equation (Equation 11) were obtained at the fifth step. However, the firing rate $r^{m}\left(t\right)$ in Equation (Equation 11) should be computed at the current iteration of the optimization.An optimization algorithm (e.g., fmincon) was run on the joint likelihood function in Equation (Equation 11) to obtain the maximum likelihood estimates of the parameters ($\theta_{ML}$ in Equation (Equation 12)).

### 3.1. Optimization Algorithm

To perform a maximum likelihood estimation (i.e., the problem defined in Equation (Equation 12)), we needed an optimizer. Most optimizers target a local minimum and thus require multiple initial guesses to increase the probability of finding a global optimum to the problem. However, this is a time consuming task and in a problem similar to that of this research duration is a crucial parameter. This was even more critical when we are using our algorithms in a physiological experiment. Some optimization algorithms such as genetic, pattern search, or simulated annealing do not require the online computation of gradients but they are computationally extensive and will most likely require a longer duration. Thus, in this research, we preferred a gradient based algorithm and utilized MATLAB’s fmincon routine. It is based on interior-point algorithms (a modified Newton’s method) and allows lower and upper bounds to be set on the result. As all parameters of an FN model are positive, a zero lower bound will prevent unnecessary parameter sweeps.

### 3.2. Simulation Scenarios

In this section, we introduce the results related to parameter estimation using table for the variation of mean estimated values $\hat{\theta }=\left[\hat{\theta_{1}},\hat{\theta_{2}},\hat{\theta_{3}},\hat{\theta_{4}},\hat{\theta_{5}}\right]$ of parameters $\theta =\left[\theta_{1},\theta_{2},\theta_{3},\theta_{4},\theta_{5}\right]$. The scenario information for the present problem appear in Table 2. To show impact of various stimulus components $N_{U}$, amplitude level $A_{\max}$, and number of trials $N_{it}$, the problem was re-run for a set of different values of those parameters.

The initial conditions of the states representing the membrane potential *V* and recovery activity *W* in Equation (Equation 1) were assumed as V(0)=0 and W(0)=0. This is a reasonable choice as we did not have any information about them.

A typical stimulus response relationship can be seen in Figure 1. Here, the stimulus parameters are $A_{\max}=100$, $f_{0}=10/3$ kHz, and $N_{U}=5$. The nominal parameters in Table 1 were used in this simulation.

### 3.3. Estimation of Parameters Using a Realistic Data

As stated in the end of the Introduction, we were likely interested in the results of the estimation when the stimulus/response data (spike trains collected) were collected from realistic neurons. Although performing an experiment may not be possible, one can use data from repositories or other sites on the web. We used the data collected in an experiment performed by de Ruyter and Bialek [27]. Here, the stimulus was of white noise type and the response was measured from H1 neurons of blowfly vision system. The data are available as a MATLAB workspace file on the website http://www.gatsby.ucl.ac.uk/~dayan/book/exercises/c1/data/c1p8.mat. In this dataset, a single stimulus of 20 min duration stimulates the H1 neurons of the flies. We divided these 1200 s long data into 2400 segments, each of which is 500 ms long. Thus, our algorithm was applied as if there were 2400 independent stimuli of 500 ms duration. Since we had a random stimulus here, we could assume that segments were triggered by independent stimuli. The algorithm was provided by subsets of data having 25, 50, 100, 200, 300, 400, 500, 600, 700, 800, 900, 1000, 1100, 1200, 1300, 1400, 1500, 1600, 1700, 1800, 1900, 2000, 2100, 2200, 2300, and 2400 samples (in other words, the value of $N_{it}$).

## 4. Results

In this section, the results of our example problem are presented. The maximum likelihood estimates $\left(\theta_{ML}\right)$ of the parameters (θ) in Equation (Equation 3) were obtained by maximizing Equation (Equation 10) using MATLAB’s fmincon routine.

The relevant results can be categorized under two headings:
The variations of mean estimated values of θ$\left(\theta_{ML}\right)$ against varying sample size $N_{it}$, amplitude level $A_{\max}$, stimulus component size $N_{U}$, and base frequency $f_{0}$ are presented in Section 4.1.The variations of standard deviations of the estimated parameters against varying sample size $N_{it}$, amplitude level $A_{\max}$, stimulus component size $N_{U}$, and base frequency $f_{0}$ are presented in Section 4.2.

### 4.1. Mean Estimated Values

One can see the variation of the mean estimated values of each parameter in Equation (Equation 3) against the number of samples $N_{it}$, amplitude $A_{\max}$, component size $N_{U}$, and base frequency $f_{0}$ of the stimulus in Table 3, Table 4, Table 5 and Table 6, respectively.

### 4.2. Standard Deviations

One can see the variation of the standard deviations of the estimates of each parameter in Equation (Equation 3) against the number of samples $N_{it}$, amplitude $A_{\max}$, component size $N_{U}$, and base frequency $f_{0}$ of the stimulus in Table 7, Table 8, Table 9 and Table 10, respectively.

In addition to the tabular results, the variation of the standard deviations are also presented in graphical forms in Figure 2, Figure 3, Figure 4 and Figure 5.

### 4.3. Results of Estimation from Realistic Data

As mentioned in Section 3.3, we also utilized realistic data obtained from H1 neurons of blowflies [27]. A little more detailed discussion is available in Section 3.3. The variation of estimated values of neuron parameters a,b,c,d,F against the sample sizes are available in Table 11. In Table 12, the relative error with respect to the case with previous sample size setting is shown. The relative error was computed with the following scheme:(13)$$ER\left(k\right)=\frac{\left(\hat{\theta }\left(k\right)-\hat{\theta }\left(k-1\right)\right)}{\hat{\theta }\left(k-1\right)}$$
where *k* refers to each of the cases in Table 11 and they are identified by the sample size parameter $N_{it}$. Here, *k* did not start from k=1 because we did not have any data concerning the cases $N_{it}<25$. Thus, in Table 12, the *k* value starts from k=2. Thus, in its first column, the relative error of the case with $N_{it}=50$ was computed against the case with $N_{it}=25$. Similarly, the relative error of the case with $N_{it}=100$ was computed against the case with $N_{it}=50$, and so on. When we examine Table 12, we can observe that the relative errors (ER) of parameters [a,b,c,d,F] reduce as the sample size increases (as *k* progresses). Although there seems a fluctuation of the relative error, the magnitude of this fluctuation tends to decrease. This is noted especially after the case with $N_{it}=600$.

### 4.4. Statistical Testing of the Parameter Estimation with Realistic Data

To test the validity of the results of Section 4.3, one needs to perform a statistical comparison test. To achieve this goal, we performed a Kolmogorov–Smirnov test on the interspike intervals of the spike trains obtained from the H1 neuron measurement data and the simulated spike trains with one of the parameter sets [a,b,c,d,F] in Table 11. As one set of measurement is not statistically adequate, we used superimposed spike sequences. As they were obtained from independent stimuli, their statistical nature was not disturbed. As we did in the estimation experiment, we superimposed the spike sequences in the response segments of both realistic measurements and the simulated output from our model. After obtaining that, we performed a Kolmogorov–Smirnov test for the two samples (one is from realistic response and one is from the simulated response from our model). We applied different segment lengths and plotted the variation of the *p*-values. The tool used in the application was MATLAB’s kstest2(x1,x2) routine (here, x1 and x2 are two samples from similar or dissimilar distributions). We used the parametric estimations from the last column in Table 11. One can see the relevant results in Figure 6, Figure 7, Figure 8, Figure 9, Figure 10 and Figure 11. From those outcomes, one can note that the *p*-value starts crossing the p=0.05 line after obtaining about 80 samples of measurement. This may be normal in the view of statistics, as these hypothesis testing algorithms require large numbers of samples to yield strong results.

## 5. Conclusions

In this paper, we present a theoretical and computational study aiming at the identification of the parameters of a single Fitzhugh–Nagumo model from stochastic discrete neural spiking data. To pursue this goal, we needed to modify the classical Fitzhugh–Nagumo model so that the output generates a firing rate instead of a membrane potential. We transformed the membrane potential information into that of a time dependent firing rate through a nonlinear map in sigmoidal form. The spiking data that are representative of an experimental application were obtained by simulating the Fitzhugh–Nagumo model and an Inhomogeneous Poisson process together. To assess the performance of the work, we repeated the simulations under different sample sizes (the number of repeated trials), stimulus component sizes, and stimulus base frequencies and amplitudes. The variation of mean estimated values and standard deviations are presented as results. The following concluding remarks can be made:The estimation algorithm showed a stable behavior for all examined conditions, as shown in Table 2.The results in Table 3, Table 4, Table 5 and Table 6 show that the mean estimated values are closest to the true values of the parameters in Table 1 when $N_{it}=100$, $N_{U}=5$, $f_{0}=0.333$ KHz, and $A_{\max}=25$.In general, the standard deviations of estimates present a decreasing behavior increasing sample size $N_{it}$ (Figure 2). For parameters *b* and *c*, there is a slightly oscillating behavior in the standard deviation values (Figure 2b,c). The standard deviations when $N_{it}=100$ are slightly larger than those of the case $N_{it}=200$. The situation may be treated inferior to the results of others studies (e.g., [8]). However, one should bear in mind that the model in [8] is a type of generic recurrent neural network and those are known to have universal approximation capabilities [33]. Thus, one should expect that the standard deviations of network weight estimates will have a better correlation to stimulus parameters when a generic model with a universal approximation capability is utilized for model fitting. In addition, the absolute standard deviations of the estimates in this research seem smaller. Thus, the overall results can be considered successful.For most of the parameters (a,b,c,d), the variation of standard deviations against the amplitude parameter $A_{\max}$ has a worsening behavior (Figure 3). The only exception is associated with the maximum firing rate parameter *F*. It has an improved standard deviation when the amplitude level $A_{\max}$ increases. Concerning the mean estimated values, changing the amplitude from $A_{\max}=25$ to $A_{\max}=200$ does not make a sensible variation. Thus, keeping $A_{\max}=25$ seems a good choice.Standard deviations of the estimates showed a little improvement when one has a large number of stimulus components $N_{U}$ (Figure 4). However, based on the mean estimated values, keeping it smaller together with the amplitude parameter $A_{\max}$ seems a viable choice.Concerning the stimulus base frequency $f_{0}$, it seems better to keep it in the lower side of the range ($0.333\leq f_{0}\leq 3.333$ KHz) applied in this research (i.e., $f_{0}\leq 1$ KHz).For assessing the performance of our model when more realistic data and longer stimuli exist, we performed an estimation attempt using the data from a previous research [27]. We divided a 20 min recording into 2400 segments, the lengths of which equal 500 ms each. The stimulus was randomly generated and thus each segment was treated as an independent experiment. It appears that the estimates of the parameters have a tendency to converge to a final value, with the increasing sample size $N_{it}$. This can be understood from the relative errors in Table 12. The errors become smaller and fluctuations diminish as the sample size advances. As a result, our model can be used in modeling studies where the computational features of the neural signal processing is important.The statistical Kolmogorov–Smirnov testing reveals that our modified Fitzhugh–Nagumo computational model can successfully describe the statistics stimulus/response relationship.

In general, the obtained results are promising. However, a slight improvement may be obtained if an optimal stimulus profile is generated prior to the identification process. The theory of optimal design of experiments [34] may be beneficial in this respect. An application to the continuous time recurrent neural network models is available in [9]. It appears to improve the mean square errors of network weight estimates (thus also the variance). This may be a part of future related studies under the same topic.

## Figures and Tables

**Figure 1 brainsci-09-00364-f001:**
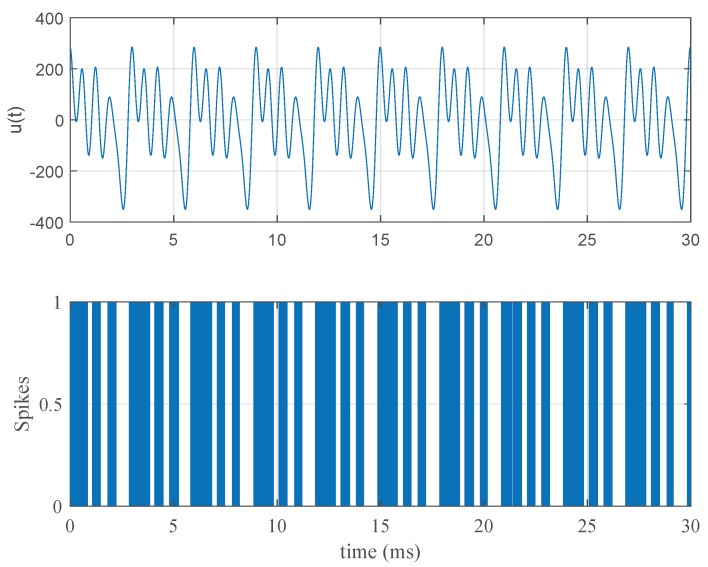
A typical stimulus and response pattern. In the first pane, a Fourier series stimulus with parameters $A_{\max}=100$, $f_{0}=333$ Hz, and $N_{U}=5$ is displayed. In the second pane, the neural spiking pattern of the Fitzhugh–Nagumo model in Equation (Equation 1) with the nominal parameters in Table 1 obtained after Poisson simulation can be seen.

**Figure 2 brainsci-09-00364-f002:**
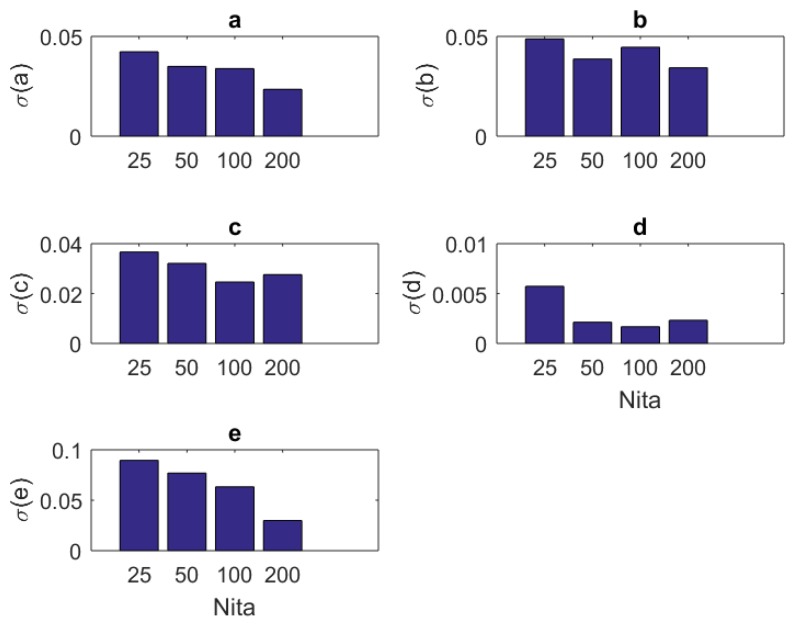
The variation of individual standard deviations (or relative errors) of the estimates against varying sample (iteration) size $N_{it}$. Other stimulus parameters are $N_{U}=5$, $A_{\max}=100$, and $f_{0}=333.3$ Hz. For most parameters, these relative errors show an improving behavior with the increasing sample size. However, some parameters such as *b* do not present any improvement or degradation in relative errors. However, in general, the relative error levels remain small.

**Figure 3 brainsci-09-00364-f003:**
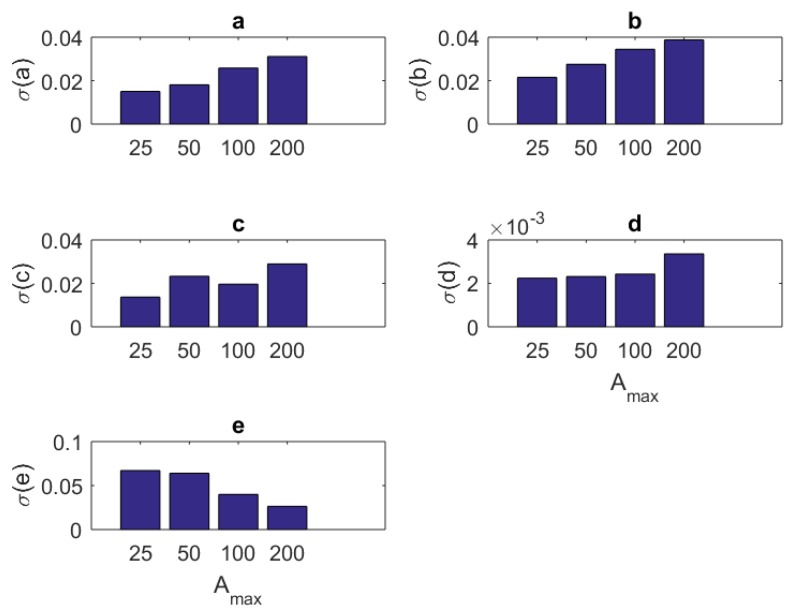
The variation of individual standard deviations (or relative errors) of the estimates against varying stimulus amplitude parameter $A_{\max}$. Other stimulus parameters are $N_{it}=100$, $N_{U}=5$, and $f_{0}=333.3$ Hz. Except for parameter *F*, one cannot see an improvement with raising the stimulus amplitude. However, in general, the relative error levels remain small.

**Figure 4 brainsci-09-00364-f004:**
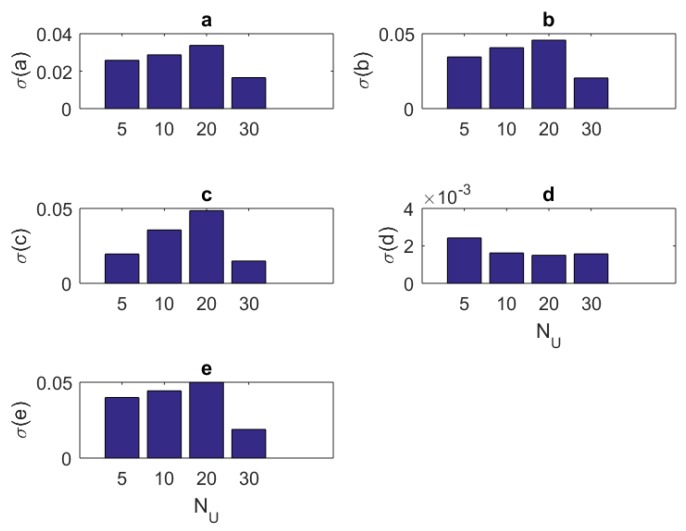
The variation of individual standard deviations (or relative errors) of the estimates against varying stimulus component size $N_{U}$. Other stimulus parameters are $N_{it}=100$, $A_{\max}=100$, and $f_{0}=333.3$ Hz. Stimuli with small $N_{U}=5$ or large $N_{U}=30$ component size can be preferred. In general, relative error levels also stay smaller in this case.

**Figure 5 brainsci-09-00364-f005:**
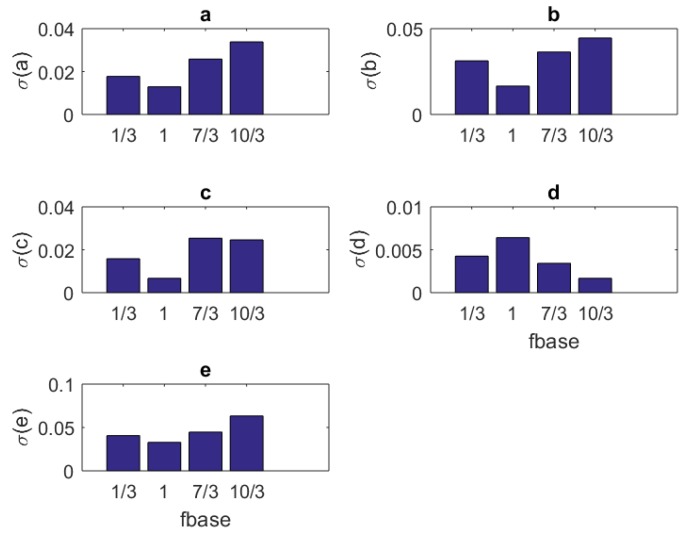
The variation of individual standard deviations (or relative errors) of the estimates against varying base frequency $f_{0}$. Other stimulus parameters are $N_{it}=100$, $A_{\max}=100$, and $N_{U}=5$. The frequencies are in KHz. Although overall relative error levels are smaller, one can prefer a mid frequency range, e.g. $1\leq f_{0}\leq MM7\;/\;3$ KHz

**Figure 6 brainsci-09-00364-f006:**
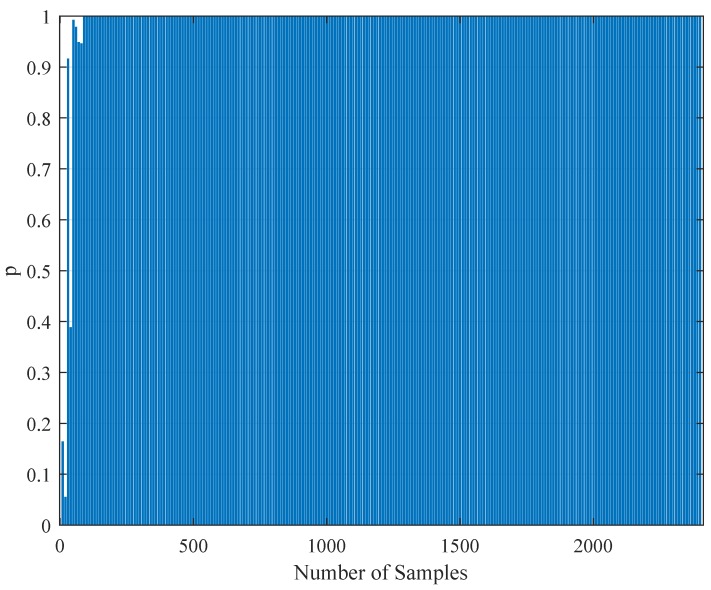
The variation of the Kolmogorov–Smirnov test *p* value with the number of samples $N_{it}$ obtained from both measurements (simulation and realistic measurement). Here, the segment size is 500 ms.

**Figure 7 brainsci-09-00364-f007:**
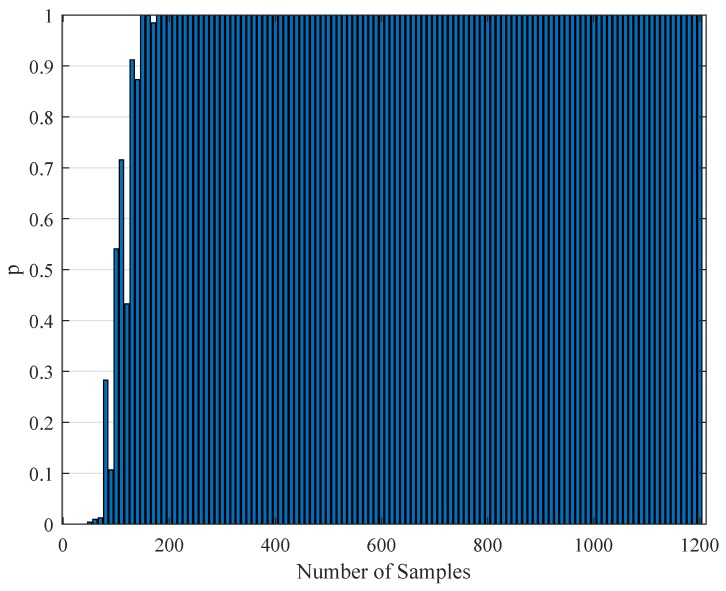
The variation of the Kolmogorov–Smirnov test *p* value with the number of samples $N_{it}$ obtained from both measurements (simulation and realistic measurement). Here, the segment size is 1 s.

**Figure 8 brainsci-09-00364-f008:**
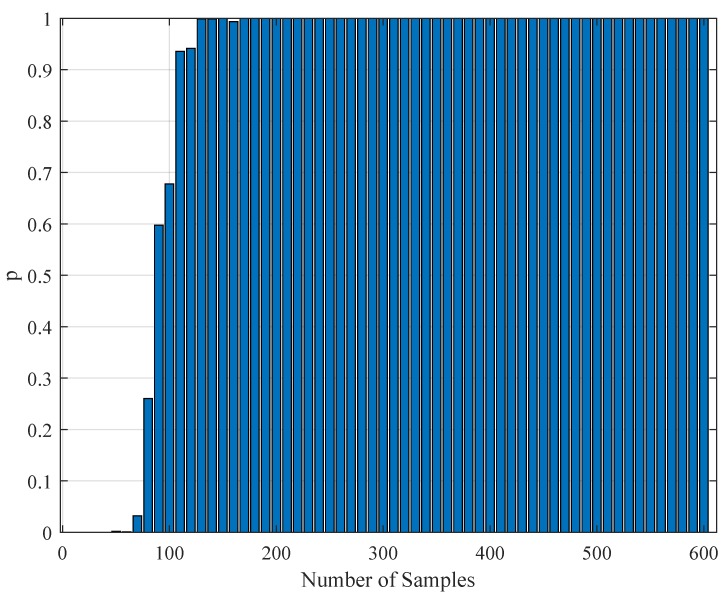
The variation of the Kolmogorov–Smirnov test *p* value with the number of samples $N_{it}$ obtained from both measurements (simulation and realistic measurement). Here, the segment size is 2 s.

**Figure 9 brainsci-09-00364-f009:**
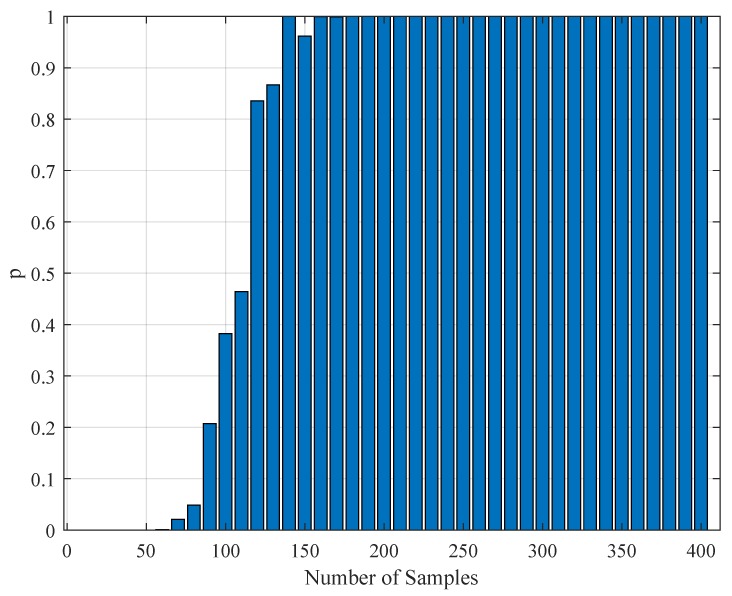
The variation of the Kolmogorov–Smirnov test *p* value with the number of samples $N_{it}$ obtained from both measurements (simulation and realistic measurement). Here, the segment size is 3 s.

**Figure 10 brainsci-09-00364-f010:**
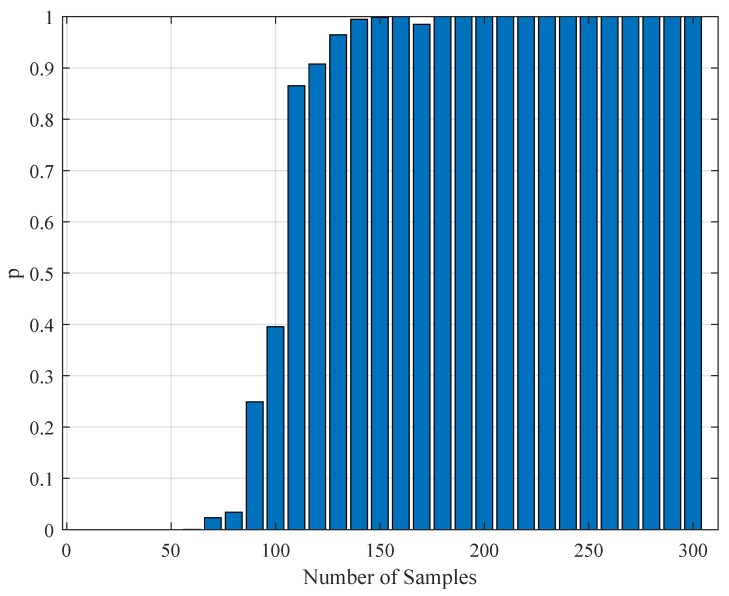
The variation of the Kolmogorov–Smirnov test *p* value with the number of samples $N_{it}$ obtained from both measurements (simulation and realistic measurement). Here, the segment size is 4 s.

**Figure 11 brainsci-09-00364-f011:**
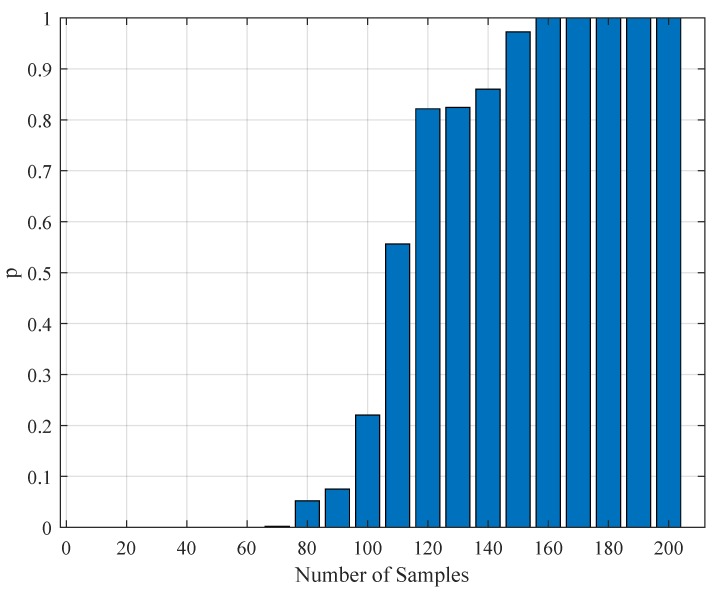
The variation of the Kolmogorov–Smirnov test *p* value with the number of samples $N_{it}$ obtained from both measurements (simulation and realistic measurement). Here, the segment size is 6 s.

**Table 1 brainsci-09-00364-t001:** The nominal parameters of the FN model in Equations (Equation 1) and (Equation 2). These were evaluated using the information in [29].

Parameter	Value
*a*	0.08
*b*	0.056
*c*	0.064
*d*	0.333
*F*	100

**Table 2 brainsci-09-00364-t002:** Data for the simulation scenario.

Parameter	Symbol	Value
Simulation Time	$T_{f}$	30 ms
Number of Trials	$N_{it}$	25, 50, 100, 200
# of Components in Stimulus	$N_{U}$	5, 10, 20, 30
Method of Optimization	N/A	Interior-Point Gradient Descent (MATLAB)
# of True Parameters	Size(θ)	5
Stimulus Amplitude (μA)	$A_{\max}$	25, 50, 100, 200
Base Frequency	$f_{0}$	$\frac{1}{3}$, 1, $\frac{7}{3}$, $\frac{10}{3}$ KHz

**Table 3 brainsci-09-00364-t003:** Estimated value vs. $N_{it}$ ($N_{U}$ = 5, $A_{\max}$ = 100, and $f_{0}=333.3$ Hz).

$N_{\mathrm{it}}$	${\hat{\theta }}_{1}$	${\hat{\theta }}_{2}$	${\hat{\theta }}_{3}$	${\hat{\theta }}_{4}$	${\hat{\theta }}_{5}$
5	0.0781	0.0504	0.0627	0.3348	100.0135
50	0.0953	0.0816	0.0731	0.3317	99.9960
100	0.0870	0.0635	0.0695	0.3326	99.9933
200	0.0840	0.0597	0.0694	0.3325	100.0065

**Table 4 brainsci-09-00364-t004:** Estimated value vs. $N_{U}$ ($N_{it}$ = 100, $A_{\max}$ = 100, and $f_{0}=333.3$ Hz).

$N_{U}$	${\hat{\theta }}_{1}$	${\hat{\theta }}_{2}$	${\hat{\theta }}_{3}$	${\hat{\theta }}_{4}$	${\hat{\theta }}_{5}$
5	0.0781	0.0504	0.0627	0.3348	100.0135
10	0.0811	0.0436	0.0595	0.3333	99.9927
20	0.0849	0.0618	0.0801	0.3326	99.9943
30	0.0770	0.0505	0.0636	0.3331	99.9920

**Table 5 brainsci-09-00364-t005:** Estimated value vs. $A_{\max}$ ($N_{it}$ = 100, $N_{U}$ = 5, and $f_{0}=333.3$ Hz).

$A_{\max}$	${\hat{\theta }}_{1}$	${\hat{\theta }}_{2}$	${\hat{\theta }}_{3}$	${\hat{\theta }}_{4}$	${\hat{\theta }}_{5}$
25	0.0817	0.0549	0.0638	0.3337	99.9980
50	0.0809	0.0586	0.0699	0.3330	100.0008
100	0.0781	0.0504	0.0627	0.3348	100.0135
200	0.0767	0.0505	0.0608	0.3322	99.9894

**Table 6 brainsci-09-00364-t006:** Estimated value vs. $f_{0}$ ($N_{it}$ = 100, $N_{U}$ = 5, and $A_{\max}$ = 100). Frequencies are in KHz.

$f_{0}$	${\hat{\theta }}_{1}$	${\hat{\theta }}_{2}$	${\hat{\theta }}_{3}$	${\hat{\theta }}_{4}$	${\hat{\theta }}_{5}$
1/3	0.0856	0.0637	0.0712	0.3315	99.9942
1	0.0796	0.0550	0.0641	0.3364	100.0124
5/3	0.0861	0.0566	0.0627	0.3327	100.0195
7/3	0.0870	0.0635	0.0695	0.3326	99.9933

**Table 7 brainsci-09-00364-t007:** Standard deviations vs. $N_{it}$ ($N_{U}$ = 5, $A_{\max}$ = 100, and $f_{0}=333.3$ Hz).

$N_{\mathrm{it}}$	$\sigma (\theta_{1})$	$\sigma (\theta_{2})$	$\sigma (\theta_{3})$	$\sigma (\theta_{4})$	$\sigma (\theta_{5})$
5	0.0423	0.0487	0.0366	0.0057	0.0895
50	0.0350	0.0387	0.0321	0.0021	0.0770
100	0.0339	0.0446	0.0246	0.0017	0.0634
200	0.0235	0.0343	0.0276	0.0023	0.02995

**Table 8 brainsci-09-00364-t008:** Standard deviations vs. $N_{U}$ ($N_{i}t$ = 100, $A_{\max}$ = 100, and $f_{0}=333.3$ Hz).

$N_{U}$	$\sigma (\theta_{1})$	$\sigma (\theta_{2})$	$\sigma (\theta_{3})$	$\sigma (\theta_{4})$	$\sigma (\theta_{5})$
5	0.0258	0.0345	0.0196	0.0024	0.0399
10	0.0287	0.0406	0.0356	0.0016	0.0444
20	0.0337	0.0457	0.0485	0.0015	0.0499
30	0.0165	0.0204	0.0149	0.0016	0.0189

**Table 9 brainsci-09-00364-t009:** Standard deviations vs. $A_{\max}$ ($N_{i}t$ = 100, $N_{U}$ = 5, and $f_{0}=333.3$ Hz).

$A_{\max}$	$\sigma (\theta_{1})$	$\sigma (\theta_{2})$	$\sigma (\theta_{3})$	$\sigma (\theta_{4})$	$\sigma (\theta_{5})$
25	0.0151	0.0216	0.0137	0.0022	0.0671
50	0.0181	0.0275	0.0232	0.0023	0.0640
100	0.0258	0.0345	0.0196	0.0024	0.0399
200	0.0311	0.0388	0.0289	0.0034	0.0264

**Table 10 brainsci-09-00364-t010:** Standard deviations vs. $f_{0}$ ($N_{i}t$ = 100, $N_{U}$ = 5, and $A_{\max}$ = 100). The frequencies are in KHz.

$f_{0}$	$\sigma (\theta_{1})$	$\sigma (\theta_{2})$	$\sigma (\theta_{3})$	$\sigma (\theta_{4})$	$\sigma (\theta_{5})$
1/3	0.0178	0.0312	0.0158	0.0043	0.0407
1	0.0129	0.0165	0.0067	0.0064	0.0329
5/3	0.0258	0.0364	0.0254	0.0034	0.0447
7/3	0.0339	0.0446	0.0246	0.0017	0.0634

**Table 11 brainsci-09-00364-t011:** The variation of estimated parameters a,b,c,d,F against increasing sample size $N_{it}$ in the estimation using realistic stimulus/response data obtained from H1 neurons of blowfly neurons.

Case #	$N_{\mathrm{it}}$	$\hat{a}$	$\hat{b}$	$\hat{c}$	$\hat{d}$	$\hat{F}$
1	25	255.7506	23.1953	344.3629	0.0000	185.6737
2	50	209.6757	21.3999	288.8835	0.0814	157.9571
3	100	233.4375	21.2668	266.9164	0.0492	154.7241
4	200	238.6861	21.1010	242.4651	0.0571	150.1093
5	300	244.5549	20.8891	239.7912	0.0777	145.6895
6	400	238.0263	20.1484	227.6343	0.1002	145.9515
7	500	220.6098	19.5167	212.1591	0.1091	142.7544
8	600	209.2398	18.9435	203.5418	0.1155	140.1229
9	700	208.3796	18.6725	200.2183	0.1180	138.9247
10	800	205.1722	18.6186	196.1978	0.1294	138.2120
11	900	206.8349	18.7251	195.6544	0.1247	137.1808
12	1000	204.2514	18.5038	192.3779	0.1250	135.9998
13	1100	201.7751	18.6313	191.4930	0.1164	136.7989
14	1200	199.1862	18.7457	190.4784	0.1237	136.2337
15	1300	196.8611	18.6375	190.3953	0.1201	135.1311
16	1400	198.3144	18.5702	190.7353	0.1230	135.3718
17	1500	196.1595	18.3109	189.0624	0.1306	134.3871
18	1600	192.2135	17.9623	185.5415	0.1447	133.7077
19	1700	190.5854	17.8516	183.7031	0.1508	133.3508
20	1800	190.7481	17.8419	184.6075	0.1495	133.5511
21	1900	192.3369	17.8900	185.2415	0.1473	133.6132
22	2000	194.9553	18.0284	185.7370	0.1495	133.5813
23	2100	198.5889	18.1381	187.4582	0.1452	134.3980
24	2200	200.3984	18.1539	188.0695	0.1366	134.8025
25	2300	201.9018	18.2673	188.5241	0.1356	134.8863
26	2400	201.6645	18.2587	187.8792	0.1357	135.2327

**Table 12 brainsci-09-00364-t012:** The relative error levels against the sample size parameter $N_{it}$. The errors were computed by evaluating the difference between the parameter values of the current case *k* and the previous case k−1 in Table 11. With increasing sample sizes, the estimates tend to have smaller fluctuations.

$N_{\mathrm{it}}$	$e_{a}$	$e_{b}$	$e_{c}$	$e_{d}$	$e_{F}$
50	0.18016	0.07741	0.16111	Inf	0.14928
100	0.11333	0.00622	0.07604	0.39521	0.02047
200	0.02248	0.00779	0.09161	0.16078	0.02983
300	0.02459	0.01004	0.01103	0.35951	0.02944
400	0.02670	0.03546	0.05070	0.28976	0.00180
500	0.07317	0.03135	0.06798	0.08886	0.02191
600	0.05154	0.02937	0.04062	0.05941	0.01843
700	0.00411	0.01431	0.01633	0.02123	0.00855
800	0.01539	0.00288	0.02008	0.09665	0.00513
900	0.00810	0.00572	0.00277	0.03619	0.00746
1000	0.01249	0.01182	0.01675	0.00208	0.00861
1100	0.01212	0.00689	0.00460	0.06885	0.00588
1200	0.01283	0.00614	0.00530	0.06300	0.00413
1300	0.01167	0.00577	0.00044	0.02875	0.00809
1400	0.00738	0.00361	0.00179	0.02359	0.00178
1500	0.01087	0.01397	0.00877	0.06198	0.00727
1600	0.02012	0.01904	0.01862	0.10819	0.00505
1700	0.00847	0.00616	0.00991	0.04207	0.00267
1800	0.00085	0.00054	0.00492	0.00871	0.00150
1900	0.00833	0.00270	0.00343	0.01443	0.00047
2000	0.01361	0.00773	0.00267	0.01429	0.00024
2100	0.01864	0.00608	0.00927	0.02861	0.00611
2200	0.00911	0.00087	0.00326	0.05905	0.00301
2300	0.00750	0.00625	0.00242	0.00735	0.00062
2400	0.00118	0.00048	0.00342	0.00091	0.00257

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
