# Peer review of "Estimating the Parameters of Fitzhugh–Nagumo Neurons from Neural Spiking Data"

_brainsci, 2019, doi:10.3390/brainsci9120364_

Round 1
Reviewer 1 Report
The manuscript entitled “Estimating the Parameters of Fitzhugh-Nagumo Neurons From Neural
Spiking Data” done by Resat Ozgur DORUK et al., utilizes the computational algorithms to measure the data and firing rate of the neuron is assumed as a nonlinear map (logistic sigmoid) relating it to the membrane potential variable is well deserved in neuroscience field.
Exploring the beneficial characteristics of computational tools applications in neuroscience might have high value the membrane potential and its bursting properties are essential in fourth order nonlinear system.
The author done the research, we will present a theoretical and computational development of an algorithm to train the parameters of a single Fitzhugh-Nagumo neuron model from a neural spiking data.
The Fitzhugh-Nagumo model represents the membrane potential of a bursting neuron. In this research, based on the knowledge the point process likelihood approaches require a firing rate information as well demonstrated. The parameters associated with this nonlinear map will also be identified in the procedure is properly described.
The methodology followed was really impressive and interesting. The authors have well utilized Fitzhugh-Nagumo (FN) model to characterize the representing the membrane potentials. The explained the neural spiking process with the mathematical model by largely obeys inhomogeneous poisson statistics, by implementing a stable poisson process simulation.
The methods were explained in a detailed and systematic manner and the corresponding results were discussed in an interactive way. The paper has been written well with clear conclusions. Beyond the potentiality of the manuscript, I have a few minor concerns that the authors may address.
Minor: Consider the more recent studies in Fitzhugh-Nagumo (FN) model and the stochastic model studies like and are more added value to this manuscript.
Author Response
Thank you very much.
REPORT: Consider the more recent studies in the Fitzhugh-Nagumo (FN) model and the stochastic model studies like and are a more added value to this manuscript.
RESPONSE: We added some comments and gave reference to four recent papers in the introduction section before introducing what is to be done in this paper. In future researches by the same authors(s), these references may provide new insights into related developments.
See the same paragraph below:
In a few research studies, Fitzhugh-Nagumo models are involved in stochastic neural spiking related studies. For example \cite{berglund2012mixed} deals with the interspike interval statistics when the original Fitzhugh-Nagumo model is modified to include noisy inputs. The number of small amplitude oscillations has a random nature and tend to have an asymptotically geometric distribution. The work by \cite{bashkirtseva2014noise} studies the effect of stochastic dynamics represented by a standard Wiener process on the limit cycle behavior. In \cite{leon2018hypoelliptic}, performs research on the hypoelliptic stochastic properties of Fitzhugh-Nagumo neurons. They studied the effect of those properties on the neural spiking behavior of Fitzhugh-Nagumo models. Finally, \cite{zhang2015parameter} investigates the stochastic resonance occurring in the Fitzhugh-Nagumo models when trichotomous noise is present in the model. It is found that when the stimulus power is not sufficient to generate firing responses, trichotomous noise itself may trigger the firing.
Reviewer 2 Report
This paper presents an approach for estimating the parameter of a single Fitzhugh-Nagumo neuron from neural spiking data information. It is well written and easy to follow, it presents the use of a sigmoid mapping to extract a firing rate from a membrane potential and then defines the use of maximum likelihood methods to estimate the parameters of the neuron. The work is interesting but has only been applied to simulated data from a single neuron and there are a number of small typographic errors.
Please expand on why the sigmoidal mapping approach is appropriate. It is not clear from the manuscript if the simulation generated spiking waveforms from only one neuron? Please clarify this point and discuss how this approach could be applied to real data where although spikes may be classified (i.e based on morphology) there will be many neurons firing at once that are off the same class. Why was the simulation run only 30 ms? This seems very short, many physiological recordings are significantly longer. How does this time scale affect the estimation of parameters? Fig. 1 to 4 are very small and the captions should be improved (made longer to explain the figures). Please add a figure that shows an example of the spiking waveform In your conclusion you mention "from noisy and discrete neural spiking data" but your work does not include any noise?Typos (by line number):
19 - remove "high" and "to"
20 - yield not yields
25 - remove "such"
75 - start the sentence with "The"
?? - where k is the number of events occur-> where k is the number of events that occur
100 - I don't understand what you mean by t and delta t become t and delta t?
167 - serach -> search
207 - that is a representative -> that is representative
Author Response
Thank you very much
Please expand on why the sigmoidal mapping approach is appropriate:
We are inspired by previous researches that relate the membrane potential and firing rate dynamics by a generic logistic sigmoid function. Some of these papers which are also cited in our paper are \cite{miller2012mathematical,doruk2018,doruk2019adaptive}. Actually, the Fitzhugh-Nagumo model here is converted into a computational model.
It is not clear from the manuscript if the simulation generated spiking waveforms from only one neuron?:
Yes, it is from a single Fitzhugh-Nagumo equation where a sigmoidal function is used to map membrane potential with the firing rate.
Please clarify this point and discuss how this approach could be applied to real data where although spikes may be classified (i.e based on morphology) there will be many neurons firing at once that are off the same class:
If the neural network is processing one dimensional signal (like the auditory cortex), one can collect the timings of the stochastic action potential bursts and apply the methodology presented in this research. We added an illustrative example in our manuscript where the data is collected by the researchers of the work experiment by de2002timing. It uses a random stimulus and recorded the spikes from H1 neurons for a duration of 20 minutes. We divided the stimulus and the related response into segments of 500 ms length and apply our approach to investigate the performance of the estimation. We present the estimated values against the sample size and show the variation of the magnitude of the error between each estimate against the sample size parameter.
Why was the simulation run only 30 ms? This seems very short, many physiological recordings are significantly longer. How does this time scale affect the estimation of parameters?
Actually, it is chosen to decrease computational duration. There seem to have some research works where such short stimuli may appear. After the proof of the success of the algorithm, one can, of course, apply longer duration stimuli. In addition, for fast-spiking neurons, one can divide a very long random stimulus into segments of very short durations and apply the algorithm developed in our research.
As we discussed above, we also prepared a simulation from real recorded spikes and stimulus. We divided the stimuli and response into 500ms segments and try whether our approach is successful or not.
It appeared that as the number of samples of 500 ms increases the estimates seem to have a converging behavior. At least the fluctuations tend to decrease.
Fig. 1 to 4 are very small and the captions should be improved (made longer to explain the figures):
Some arrangements are made. Scale enlarged. Captions now include comments on the results.
Please add a figure that shows an example of the spiking waveform:
It is being prepared.
In your conclusion you mention "from noisy and discrete neural spiking data" but your work does not include any noise?:
By meaning noisy, we meant the stochasticity of the neural spikes but we admit that it is quite inappropriate here. So we changed it with the term "stochastic".
Typos (by line number): Corrected as much as we can.
Reviewer 3 Report
This manuscript describes a method to determine parameters of a Fitzhugh-Nagumo (FN) neuronal model by fitting spike time, rather than amplitudes or other variables from experimental (here simulated) data. Standard maximum likelihood methods are employed to perform the fits. Key to the proposed method is derivation of spike times from the nonspiking FN model by a threshold test and fitting of those spike times to an inhomogeneous Poisson process to obtain likelihood density for the parameter fitting process. The methodology appears sound and probably even useful given the frequent availability of spike-time data from neuronal recordings.
My problem with this manuscript is that I am not convinced that the numerical tests are adequate to make the case that the proposed method will be generally useful for different neuronal cell types. Specifically, as far as I can understand the presentation, all the test data were derived from a single example of an FN model using parameters given in Table 1. Multiple stimulus sets were generated from this model and tested, varying the parameters listed in Table 2, but at no point was a demonstration made that the FN parameters (a,b,c,d,F in Table 1) could be varied and then recovered from simulated data that might be obtained from cells operating according to those parameters.
Even more desirable would be to take data from real cell recordings, many examples of which are freely available on the Internet (see, for example, https://crcns.org). It is understood that in this case, correct values of the FN parameters are not known, however, I would suggest that FN parameters could be derived for a few of these data sets by the proposed methods. Then the test would be whether simulated data derived from the FN fits would fit the same interspike interval distribution as the data used to generate the fit. If not, of course, one could discuss reasons why the particular data set does or does not represent data that can be well fit by an FN model and whether more recent or more complex models (e.g. Izhikevich spiking neuron model) might provide better fits. (I realize that this suggestion goes beyond the scope of the present ms. I would urge that the authors consider doing this type of test of their model, as it would make the paper much more convincing to a reader versed in computational neuroscience, but I do not consider this a condition of acceptance if the test described above with different a,b,c,d,F is included.)
I have the following additional comments:
I would like to see a little more justification for use of a sigmoid relation of membrane potential to spike rate, i.e. mention situations where this might not be valid. Similarly, a little more discussion of why the cosine Fourier series with random phase was chosen as a model for stimulus generation.
Standard deviations shown in Tables 7-10 would be more understandable if restated as relative (i.e. fractional) errors in the respective parameters.
Line 100 has this strange sentence: "and so t and delta-t become t and delta-t". I think there is some typo here.
In Eqn. 7, r-sub-e is not defined or explained.
English could be improved.
Author Response
Thanks for suggestions.
We now have an attempt to estimate the parameters a,b,c,d,F from realistic data. The data is from previous research which applies a random stimulus to the vision system of blowflies. The spike timings are collected from the H1 neurons and stored as a MATLAB .mat file. The 20 minutes recording is divided into 2400 segments of which are 500ms each.
Since the stimulus is of white noise type, each segment can be treated as an independent experiment with random stimuli. So our algorithm can be applied successfully without extensive changes.
Concerning the Fourier Series stimulus, we choose that because of the principal author's previous works. There are a bunch of papers that the analysis and estimation involve phased cosine Fourier series. In addition, one can advance this work to implement an optimal experiment design which involves optimization of the stimulus parameters based on an information metric. So a stimulus like a truncated Fourier series may be a good choice because it has a number of well-defined parameters that can be utilized in optimization.
Also in auditory neuroscience research Fourier series may be a viable choice to represent sound stimuli.
The choice for a sigmoidal relationship that maps the membrane potential and firing rate is also coming from our experience in previous research.
We added related comments in the manuscript. These discuss our references which inspire us in those choices.
In Eqn. 7, r-sub-e: is a typo sorry for that.
The other typo is also corrected.
Round 2
Reviewer 3 Report
The authors have provided satistfactory modifications in response to my minor comments on their ms. However, they still did not provide numerical tests adequate to make the case that the proposed method will be generally useful for different neuronal cell types. I suggested two ways to do this:
(1) To generate simulated data sets from FN models with different parameters (a,b,c,d,F), apply their method to these simulated data sets, and see how well the (a,b,c,d,F) parameters would be recovered. They chose not to take this approach.
(2) To apply the method to data from real cell recordings. I was very pleased that the authors took this approach and were able to provide results very quickly and with apparent success. However, they did nothing to show that the FN parameters obtained provide a reasonable representation of the input neuronal spiking signal. Rather, they only provide data to show that standard deviations in the obtained FN parameters are reduced by inclusion of more spikes (longer recording times) in the input data. This is reassuring, but does not answer the question. As I suggested earlier, they need to generate simulated spike trains using the FN parameters obtained from their new analysis of real data (they already have software to do this that was used in the original study) and show by some statistical test that these have the same distribution as the input data to within some error. While I do not wish to specify exactly which test should be used here, I would suggest that the one-dimensional Kolmogorov-Smirnov test provides a suitable nonparametric test for identity of distributions that can be applied to noncontinuous data such as distributions of interspike intervals.
The need for this additional test seems particularly acute to this reviewer in that the a,b,c,d,F parameters shown in the new Table 11 a larger in the cases of a,b,c by several orders of magnitude from the "standard" values shown in Table 1. This is worrying because these three parameters together provide the value of W-dot and W-dot is subtracted from V-dot in Eqn. 1, which would seem to lead to a very rapid instability in the time course of the model voltage. Perhaps there was an error in the preparation of Table 11, or perhaps the maximum likelihood solution found an incorrect local maximum in the parameter space.
Also, Tables 11 and 12 are longer then necessary; perhaps a selection of five or six pairs of cases taken from across the full range of Nit values would be sufficient.
Line 254: "In section we stated..." the section number is missing.
Author Response
We provided a Kolmogorov-Smirnov based analysis at the end of the last section. We tried superimposed sequences of spikes from both realistic and simulated data. We tried different lengths of segments and after having enough number of samples they have p-values larger than 0.05.
Besat regards,